# ONE IS MORE: DIVERSE PERSPECTIVES WITHIN A SINGLE NETWORK FOR EFFICIENT DRL

## ABSTRACT

Deep reinforcement learning has achieved remarkable performance in various domains by leveraging deep neural networks for approximating value functions and policies. However, using neural networks to approximate value functions or policy functions still faces challenges, including low sample efficiency and overfitting. In this paper, we introduce OMNet, a novel learning paradigm utilizing multiple subnetworks within a single network, offering diverse outputs efficiently. We provide a systematic pipeline, including initialization, training, and sampling with OMNet. OMNet can be easily applied to various deep reinforcement learning algorithms with minimal additional overhead. Through comprehensive evaluations conducted on MuJoCo benchmark, our findings highlight OMNet's ability to strike an effective balance between performance and computational cost.

## 1 INTRODUCTION

Deep reinforcement learning, as evidenced by various studies (Mnih et al., 2015; Silver et al., 2017; Jumper et al., 2021), has demonstrated its versatility across numerous domains, surpassing the scope of traditional reinforcement learning. This achievement can be attributed to the remarkable function approximation prowess exhibited by deep neural networks. Within the realm of deep reinforcement learning, neural networks play a pivotal role in approximating value functions, policy functions, and other critical components.

However, recent research has pointed out that training neural networks in deep reinforcement learning still faces several challenges (Igl et al., 2020; Nikishin et al., 2022; Lyle et al., 2022; 2023; Sokar et al., 2023), which, in turn, affect the performance of deep reinforcement learning agent. One prominent issue among these challenges is the overfitting of neural networks in deep reinforcement learning algorithms. Neural networks may be prone to overfitting on early experiences (Igl et al., 2020; Nikishin et al., 2022; D'Oro et al., 2022) or bootstrapped learning target (Lyle et al., 2022; 2023) in deep reinforcement learning, resulting in poor sample efficiency (Nikishin et al., 2022) or generalization performance (Igl et al., 2020).

In the face of the difficulties mentioned above, one class of methods involves the use of ensemble learning techniques in deep reinforcement learning. Ensemble learning has found widespread application in deep reinforcement learning. For example, researchers employ multiple value networks to enhance value estimation in deep reinforcement learning, addressing issues such as alleviating overestimation (Fujimoto et al., 2018), reducing variance in value estimates (Anschel et al., 2017; Lee et al., 2021), and avoiding overfitting of value networks (Chen et al., 2021), among others. Similarly, maintaining multiple policy networks helps achieve more diverse behaviors, which can improve sampling efficiency (Zhang & Yao, 2019; Lee et al., 2021; Yang et al., 2022; Fan et al., 2023) or enhance an agent's generalization capabilities (Yang et al., 2022) in complex and diverse environments. However, ensemble learning comes at the cost of increased parameter complexity and additional training overhead. In ensemble reinforcement learning, it often takes several times the training effort to achieve performance gains, imposing severe constraints on the practical application of such algorithms.

Different from employing extensive network ensembles, our inspiration stems from biological facts, which offer an alternative perspective on addressing the issue. Research in the field of biology indicates that sparsity plays a crucial role in biological neural networks (Friston, 2008; Foldiak, 2003; Herculano-Houzel et al., 2010). This implies that when the brain responds to a specific problem,

only a fraction of the neural connections are truly active. Similar to biological neural networks, researchers have found that the concept of sparsity can also be explored and utilized in deep neural networks. Recent research on sparse neural networks has demonstrated their impressive performance (Han et al., 2015; Frankle & Carbin, 2018; Liu et al., 2017; Malach et al., 2020). The "lottery ticket hypothesis" (Frankle & Carbin, 2018) suggests that even in cases of very limited parameters, a sparse network can be trained to achieve performance comparable to dense neural networks. In the field of deep reinforcement learning, there is also substantial work highlighting the potential of sparse neural networks (Sokar et al., 2021; Graesser et al., 2022; Tan et al., 2022; Arnob et al., 2021). These studies imply that it is possible to harness untapped potential in neural networks even without increasing the number of parameters.

Inspired by the aforementioned ideas, we pose the following question: *Is it possible to achieve diverse perspectives using just a single neural network to enable efficient deep reinforcement learning?* More specifically, we aim to find a method that allows the network to generate more diverse outputs (such as value estimates produced by the value network or actions generated by the policy network) without increasing the network parameters. This is to achieve a more effective and efficient deep reinforcement learning algorithm. To address this question, we focus on a single neural network and discover that a single neural network can be transformed into multiple overlapping subnetworks, allowing for more diverse network outputs without increasing parameter complexity or computational overhead. We refer to our approach as OMNet, named after "One is More."

Our contributions are as follows:

- We show that one can leverage the presence of multiple distinct subnetworks within a single neural network, which provides diverse outputs that can be harnessed. We refer to this algorithm as OMNet and have introduced the algorithm pipeline for initializing, training, and utilizing OMNet for sampling.

- We applied OMNet to deep reinforcement learning algorithms and obtained an efficient reinforcement learning approach that balances both sampling efficiency and computational efficiency. We show that using OMNet as the value network efficiently yields more accurate value estimates, alleviating the issue of value overestimation without the need for maintaining multiple value networks. We also demonstrate that OMNet can be used as the policy network to enable the collecting of richer and more diverse trajectories, accelerating the exploration of different states in the early stages of training.

- We validate our method in continuous control environments, and the experiments demonstrate that OMNet can achieve both high sampling efficiency (converging performance within 300K environment steps on MuJoCo) and computational efficiency (over 5 times reduction in computation compared to the state-of-the-art algorithm). Furthermore, through ablation experiments, we establish the stability of OMNet with respect to hyperparameter choices, making this algorithm easy to implement and use.

## 2 RELATED WORKS

**Sample Efficiency**  Efforts to improve sampling efficiency in deep reinforcement learning have been pivotal for practical, real-world applications. Model-based reinforcement learning methods, such as Yu et al. (2020); Janner et al. (2019); Schrittwieser et al. (2020); Ye et al. (2021), strive to bolster agent planning by training a transition model. However, the costs associated with this model's training and its impact on agent planning accuracy are notable challenges. Recently, model-free reinforcement learning algorithms have shown comparable or superior sampling efficiency, particularly by increasing the replay ratio, as discussed by D'Oro et al. (2022). In continuous control, several approaches employ techniques such as model ensembles (Chen et al., 2021) or modified model architectures (Hiraoka et al., 2021) to mitigate overfitting and achieve sampling efficiency akin to state-of-the-art model-based methods. Additionally, researchers have observed that periodically resetting deep reinforcement learning model parameters can lead to substantial gains in sampling efficiency (Nikishin et al., 2022; D'Oro et al., 2022).

**Sparsity in NN**  People have studied the sparsity of deep neural networks from various perspectives. Researchers use Dropout (Srivastava et al., 2014) or DropConnect (Wan et al., 2013) during the training process to sparsely mask the parameters of the neural network. This approach has been shown to be beneficial in improving the generalization capability of neural networks. Previous works on pruning (Han et al., 2015; Lee et al., 2018; Wang et al., 2020) and "Lottery Ticket Hypothesis"

(Frankle & Carbin, 2018; Frankle et al., 2020; Evci et al., 2020) have demonstrated the role of sparse neural networks in efficient training and deployment. In deep reinforcement learning, a recent line of work (Sokar et al., 2021; Graesser et al., 2022; Tan et al., 2022; Arnob et al., 2021) suggested that a sparse subnetwork of a dense network can preserve performance with only a small fraction of parameters. The above work reveals the potential of neural networks with a limited number of parameters.

**Ensemble Methods in DRL**  In deep reinforcement learning, ensemble learning methods are pivotal. For example, TD3 (Fujimoto et al., 2018) in continuous control employs Clipped Double Q-learning as a form of ensemble learning with an ensemble size of 2 to address value network overestimation. To enhance value estimation stability in online and offline settings, researchers have utilized larger ensembles for value networks (Anschel et al., 2017; Agarwal et al., 2020; Lee et al., 2021). REDQ (Chen et al., 2021) employs a stochastic ensemble approach to reduce learning bias in value networks, particularly in high replay ratio scenarios, thereby improving sample efficiency. Some studies also apply ensemble methods to policy networks to enhance agent exploration (Zhang & Yao, 2019; Lee et al., 2021). In model-based algorithms, ensemble techniques are employed to enhance the learning of environment transition dynamics and dynamic model accuracy (Kurutach et al., 2018; Buckman et al., 2018; Yu et al., 2020). These ensemble approaches contribute to the robustness and effectiveness of reinforcement learning agents in DRL.

## 3 OMNET: ONE IS MORE

In this section, we present the idea and design details of our OMNet method. Our approach is inspired by existing research on sparse networks, which demonstrates that performance comparable to a dense neural network can be achieved by training or inferring with a sparse subnetwork within a neural network (Han et al., 2015; Frankle & Carbin, 2018; Liu et al., 2017; Malach et al., 2020). However, differing from existing pruning or sparse training algorithms, our approach, instead of reducing the neural network's parameter count through sparsity, focuses on maintaining multiple subnetworks within the network during training without altering the overall parameter count. The goal is to enable the neural network to produce more diverse outputs with minimal additional computational cost. In the remaining part of this section, we will sequentially introduce how we initialize and train an OMNet and utilize OMNet for sampling.

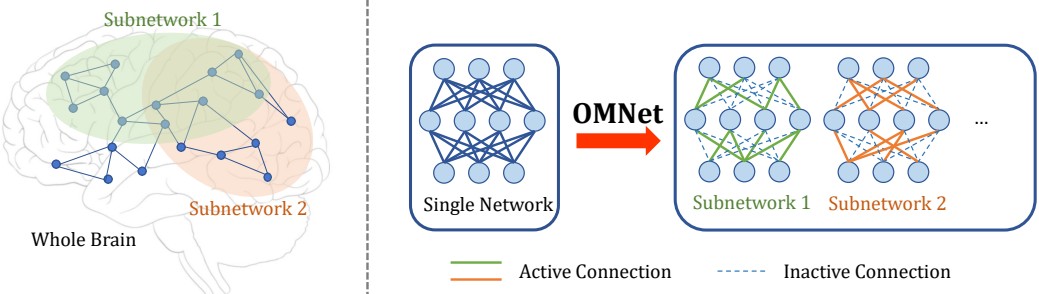

Figure 1: Overview of OMNet. Analogous to the sparsity in the biological brain, we highlight that multiple diverse subnetworks can coexist within a single neural network, i.e., "One is More." OMNet can yield improvements in the sampling efficiency, computational efficiency, and generalization of deep reinforcement learning algorithms.

### 3.1 INITIALIZATION OF OMNET

We first describe how we obtain a subnetwork from a given neural network through the use of a binary mask.

Specifically, given a neural network parameterized by $\theta \in \mathbb{R}^n$, where $n$ refers to the number of parameters, the parameter of a subnetwork is $\theta \odot m$, where $\odot$ is the element-wise product and $m$ is the corresponding binary mask $m \in \{0,1\}^n$. This concept has been referenced in many prior works, e.g., Frankle & Carbin (2018); Frankle et al. (2020); Cosentino et al. (2019); Jaiswal et al. (2022).

Unlike previous pruning (Han et al., 2015; Liu et al., 2017; Zhu & Gupta, 2017; Dong et al., 2017) or sparse training (Mocanu et al., 2018; Lee et al., 2018; Evci et al., 2020) methods, that only maintain a single binary mask, our method maintains a set of fixed masks during training, i.e., $\mathcal{M} = \{m_1, m_2, \cdots, m_N\}$, where $N$ is the number of masks. In this way, we get access to multiple subnetworks $\theta_i = \theta \odot m_i$ with almost no extra overhead since the binary masks are freezed, i.e., they do not need to be tracked or have gradients computed during training, and can be easily stored compared to floating-point format parameters. For each binary mask, we independently sample from the same Bernoulli distribution, which takes value 1 with probability $1 - S$ and value 0 with probability $S$, and these masks remain unchanged throughout the subsequent training process. This generation approach has very low complexty, and we will see later that it is also highly effective.

## 3.2 TRAINING AN OMNET

Given the set of subnetworks generated, we now describe the training procedure. Consider a general neural network denoted by $f_\theta(x)$, where $\theta$ is the parameter and $x$ is the input. We can express its loss function by

$$\mathcal{L}(\theta) = \ell(\theta, f_\theta(x)). \tag{1}$$

To implement an OMNet, we modify the above loss function to

$$\mathcal{L}(\theta_i) = \hat{\ell}(\theta, f_{\theta_i}(x)), i \sim \mathcal{U}[1, N] \tag{2}$$

Here $i \sim \mathcal{U}[1, N]$ means we choose an $i$ uniformly at random in $\{1, 2, ..., N\}$. From Eq. (2), we see that there are two key points in training an OMNet:

(i) We use a modified loss function $\hat{\ell}(\theta, y)$. Since we get access to multiple subnetworks, we can make use of the methods in ensemble training, for instance, using the average loss of all networks as $\hat{\ell}(\theta, y) = \frac{1}{N} \sum_{k=1}^{N} \ell(\theta_k, y)$. Similarly, other ensemble-based methods designed to enhance deep reinforcement learning algorithms can be applied to OMNet in a similar manner. The key adaptation involves replacing the operators originally designed to operate on multiple distinct networks with operations performed on multiple subnetworks within OMNet.

(ii) In each iteration, we only update one of the subnetworks in OMNet with the random index $i$. This approach is different from previous works on ensemble training, which requires training all networks in parallel and results in a rapidly increasing training overhead as the number of ensemble members grows (Mohammed & Kora, 2023). In contrast, as the parameters overlap among the subnetworks trained in OMNet, the number of parameters updated in each iteration is only influenced by the sparsity of the subnetwork and is independent of the number of subnetworks. This allows us to obtain multiple well-performing subnetworks efficiently by updating only one subnetwork at a time in each iteration.

Based on the criteria mentioned above, OMNet can be straightforwardly applied to various deep reinforcement learning algorithms. In this paper, we experimented with combining OMNet with the state-of-the-art SAC algorithm in the domain of continuous control. For more details about the SAC algorithm and how we integrated OMNet with SAC, please refer to Appendix A.

## 3.3 SAMPLING WITH AN OMNET

Many studies have pointed out that maintaining multiple policies can help the agent exhibit more diverse behaviors (Zhang & Yao, 2019; Lee et al., 2021; Yang et al., 2022; Fan et al., 2023). In this paper, we employ a simple strategy to leverage different subnetworks within the actor for sampling. Specifically, for each episode, we randomly select one subnetwork from the actor to make decisions, and we maintain the use of the same subnetwork throughout each episode. This approach helps diversify the sampling trajectories of different subnetworks, enabling the agent to collect more diverse data. We will demonstrate the advantages of using OMNet for sampling in Section 4.3 and present visualization results.

## 4 EXPERIMENTS

In this section, we conduct a comprehensive set of experiments to demonstrate the effectiveness of OMNet. In Section 4.1, we compare OMNet with the current state-of-the-art baseline algorithms, showing that OMNet can achieve a balance between computational efficiency and sampling efficiency. It achieves equal or higher sampling efficiency with much fewer parameters and lower computational overhead. In Section 4.2, we delve deeper into how OMNet improves both value networks and policy networks. We show that OMNet enables more accurate value estimates from the value network and increases exploration in the policy network. In Section 4.4, we introduce environmental noise to investigate OMNet's generalization performance, confirming that OMNet can

enhance the algorithm's generalization capabilities. In Section 4.5, we conduct an ablation study to analyze the sensitivity of OMNet to sparsity and the number of subnetworks.

## 4.1 OMNET ENABLES HIGH SAMPLE EFFICIENCY AND COMPUTATIONAL EFFICIENCY

We conduct experiments of OMNet on four tasks from MuJoCo: Hopper-v4, Walker2d-v4, Ant-v4, and Humanoid-v4, as the same as previous works on sample efficiency on MuJoCo benchmark (Chen et al., 2021; Hiraoka et al., 2021; Li et al., 2023). Each experiment is repeated 10 times with different random seeds, and we report the average result.

We consider the following methods: (i) **SAC** (Haarnoja et al., 2018): the original SAC algorithm, updating the network parameters once after each environment interaction. (ii) **SAC-20**: SAC with replay ratio (D'Oro et al., 2022) of 20, i.e., updating the network parameters for 20 times after each environment interaction. (iii) **REDQ** (Chen et al., 2021): REDQ uses a replay ratio of 20 and policy delay and trains 10 value networks in parallel while calculating the target Q value with a random subset of value networks. (iv) **DroQ** (Hiraoka et al., 2021): Similar to SAC, with a replay ratio of 20 and a different value network architecture by using dropout (Srivastava et al., 2014) and layer normalization (Ba et al., 2016). (v) **SR-SAC** (D'Oro et al., 2022): SR-SAC uses SAC with a high replay ratio and resets the network parameters periodically. In our implementation, we reset the parameters every $10^5$ environment step. (vi) **OMNet**: Our method, using OMNets for both value network and policy network. We also adopt layer normalization in network architectures. We emphasize that OMNet can generate diverse outputs using just a single neural network. Therefore, unlike all previous algorithms, we employ only one value network, calculating the TD target by taking the minimum value estimate from two different subnetworks. In the implementation of OMNet, we employed 5 subnetworks with a sparsity of 0.5 for all environments. In the rest of Section 4, unless otherwise specified, we used the same set of hyperparameter settings for OMNet. All methods except for SAC use a replay ratio of 20 and policy delay to keep consistency.

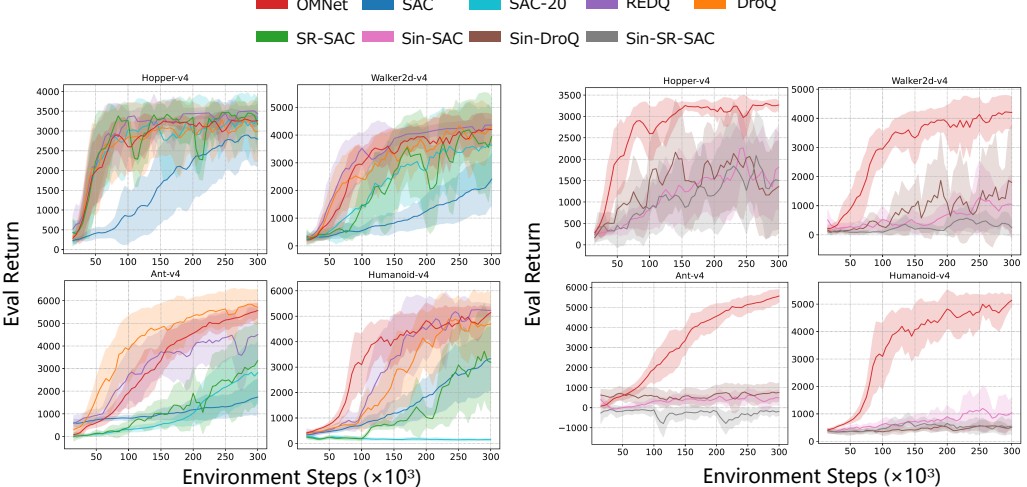

(a) Comparison with algorithms with multiple value networks.

(b) Comparison with algorithms with single value network.

Figure 2: Training curves of 300K environment steps on MuJoCo. In Figure 2(a), we compare OMNet with existing state-of-the-art high-sample-efficiency algorithms. In Figure 2(b), we compare OMNet with variants of the baseline algorithms that use only a single value network. OMNet can strike a balance between sample efficiency and computational efficiency, achieving optimal performance on both dimensions.

The training curves are presented in Figure 2. We run each method on each environment with $3 \cdot 10^5$ environment steps. Figure 2(a) shows the training curves in comparison with baseline algorithms. REDQ utilizes 10 value networks, and the other baseline algorithms use 2 value networks each, while OMNet adopts only one value network. For DroQ, we optimize its dropout ratio by searching over a large range. Note the performance of DroQ can be sensitive to the dropout rate. Training curves and the selection of dropout rate for each environment are provided in Appendix C. OMNet shows comparable performance with REDQ by effectively utilizing different subnetworks within a single neural network, without the need for a large ensemble of value networks as in REDQ. Specifically, in the complex control environment of Humanoid-v4 with 376-dimensional inputs,

OMNet exhibits superior sampling efficiency compared to various baseline algorithms. Moreover, our ablation study in Section 4.5 shows that OMNet exhibits stable performance across different hyperparameter settings, thus delivering performance improvements without the need for meticulous hyperparameter tuning.

To compare fairly with baseline algorithms in the case of using only one value network, we further conducted a comparison with variants of baseline algorithms that utilize a single value network. We consider the following baselines with a single value network: (i) **Sin-SAC**: uses only one value network in SAC; (ii) **Sin-DroQ**: the single-network version of DroQ, as originally proposed in Hiraoka et al. (2021); (iii) **SR-Sin-SAC**: Sin-SAC with network reset. For Sin-DroQ, we meticulously selected dropout rates for each environment individually and included the training curves and the chosen dropout rates for each environment in Appendix C. Figure 2(b) illustrates the training curves of OMNet and baseline algorithms. It can be observed that the baseline algorithms fail to achieve satisfactory performance when using only one value network.

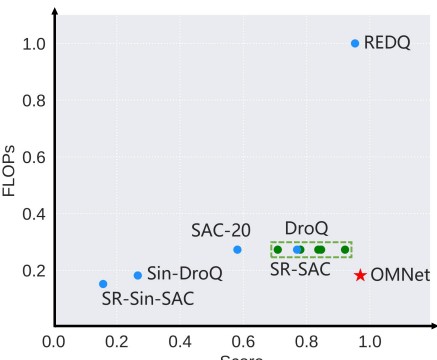

In Figure 3, we also present an efficiency comparison of different approaches. The horizontal axis corresponds to the training FLOPs of the algorithms, which represents the number of FLOPs required for one iteration of all networks during the training process, normalized relative to the FLOPs of REDQ. The vertical axis provides the scores of the algorithms. We adopt the average evaluation returns from the last 50k environment steps of experiments conducted with a single run as the score for that run. Then, we normalize the returns by the best average return achieved on each environment, as used in Li et al. (2023), and compute the average results across all seeds and environments as the algorithm's score. The green dots within the green dashed box represent the performance of DroQ at different dropout rates. For Sin-DroQ, we only plotted the performance corresponding to the best dropout rate. It can be observed that OMNet achieves an optimal bal-

Figure 3: Comparison of training efficiency and sampling efficiency among different algorithms. The green dots represent the performance for DroQ at different dropout rates.

ance between high sampling efficiency and computational efficiency, surpassing existing algorithms. It is worth noting that the performance of DroQ is sensitive to the choice of dropout rate, as shown in Appendix C. In contrast, in Section 4.5, our ablation experiments on OMNet demonstrated that OMNet's performance is stable across hyperparameter values.

## 4.2 OMNET IMPROVES VALUE ESTIMATION

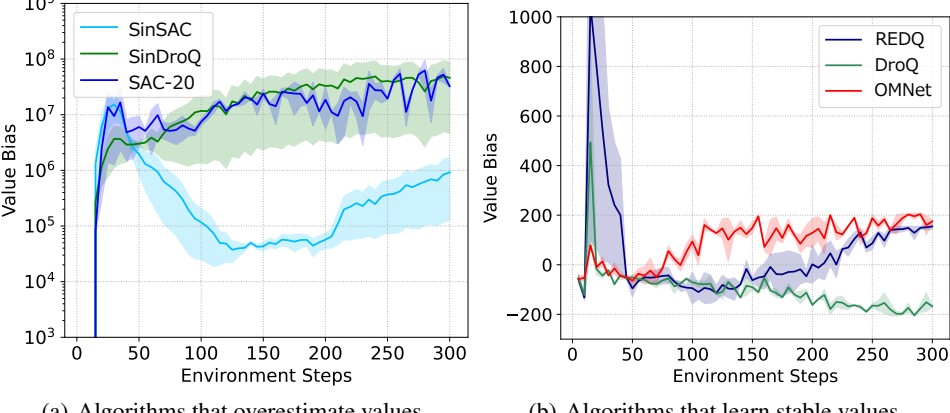

(a) Algorithms that overestimate values.     (b) Algorithms that learn stable values.

Figure 4: Value estimation bias during the training process for different algorithms. To display the results on an appropriate scale, we present the curves for algorithms where value overestimation occurs significantly in Figure 4(a), and in Figure 4(b), we show the curves for algorithms with more stable value estimates. OMNet is capable of learning smaller value estimation biases even when using only one value network.

As demonstrated in Section 4.1, OMNet achieves strong performance even when using just one value network, a feat not matched by other baseline algorithms. To further elucidate how OMNet enhances better learning of the value network, we calculate the bias of the value network estimates during the training process, i.e., $\mathbb{E}_{\pi}[(Q_{\theta}(s,a) - Q^{\pi}(s,a))]$. We calculate the value bias of Sin-SAC, Sin-DroQ, SAC-20, REDQ, DroQ, and OMNet, and the results are displayed in Figure 4. These experiments were conducted on the Humanoid-v4 environment.

In Figure 4(a), we illustrate three cases of value overestimation, where we observe that these algorithms significantly overestimate values (up to $10^7$), which corresponds to very poor performance by these algorithms as shown in Section 4.1. This indicates that applying a high replay ratio to the SAC algorithm or using a baseline algorithm with only one value network leads to significant value estimation overestimation. On the other hand, in Figure 4(b), we present the curves depicting the change in value bias over sampling steps for REDQ, DroQ, and OMNet. For DroQ, we utilized a fine-tuned dropout rate. It can be observed that all three algorithms eventually learn relatively stable values, with OMNet maintaining a small value bias throughout the entire training process. It is noteworthy that among these three algorithms, REDQ employs 10 value networks, DroQ uses 2 value networks with carefully chosen dropout rates, while OMNet leverages multiple subnetworks within a single value network (hence much fewer parameters and lower complexity). These results demonstrate the effectiveness and efficiency of OMNet in improving value estimation.

### 4.3 OMNET COLLECTS DIVERSE TRAJECTORIES

In deep reinforcement learning, agents need to interact with the environment continuously to collect data, and the quality of the data collected can significantly impact the training of neural networks, thereby having a crucial effect on algorithm performance. In this section, we investigate the impact of using OMNet as the policy network and empirically demonstrate that using OMNet as the policy network helps the agent collect more diverse trajectories. To visually demonstrate the effects of OMNet, we consider a simple sparse-reward 2D maze environment, as depicted in Figure 5(a). The maze is a $1 \times 1$ square, and the agent always starts from the center of the maze. The agent receives rewards when it approaches the destination near the upper-right corner of the maze. Please refer to Appendix B.1 for more details about this environment.

We compare two different agents: one that uses OMNet for both the value network and policy network, the same as the agent evaluated in Section 4.1, and the other that uses OMNet solely for the value network. We start by comparing their sample efficiency based on the average evaluation return. As shown in Figure 5(b), using OMNet as the policy network leads to higher sampling efficiency. This indicates that employing OMNet can improve the performance of the policy network.

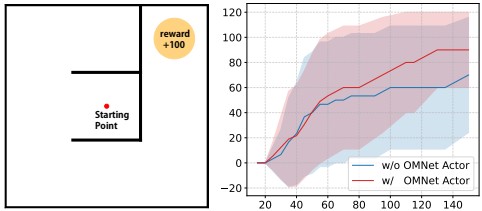

(a) Map of 2D maze. (b) Training curves in 2D maze.

Figure 5: An environment where an agent navigates through a 2D maze. The agent starts from the center of the maze and needs to reach the target point located in the upper-right corner while navigating around obstacles. When using OMNet as the actor, the agent exhibits higher sampling efficiency.

To compare the impact of using OMNet as the policy network on the sampling trajectories, we visualize the exploration of different parts of the map by the agent in the early stages of training. The experimental results are averaged over multiple repeated experiments. The red area indicates regions explored by the agent, with darker red representing higher visitation frequencies. Further details of the visualization of these results can be found in Appendix B.2.

It can be seen that within just 100 steps of interaction with the environment, most of the exploration by agents not using OMNet as the policy network remains concentrated in a small area within the three-sided wall in the center of the maze. This implies that a significant portion of the exploration steps are spent bumping into walls and taking detours within this small area. On the other hand, agents using OMNet as the policy network cover a larger portion of the maze on the left and top within the first 100 steps, and there are several visits to the right side of the maze, near the ultimate goal location. This indicates that using OMNet as the policy network covers more areas within the same 100 exploration steps. The visualization results for 500 and 1000 steps also support the conclusion that using OMNet as the policy network allows for faster coverage of more areas of the map within the same number of sampling steps. This result highlights the improvement that using OMNet as the policy network can bring to the early exploration of the agent.

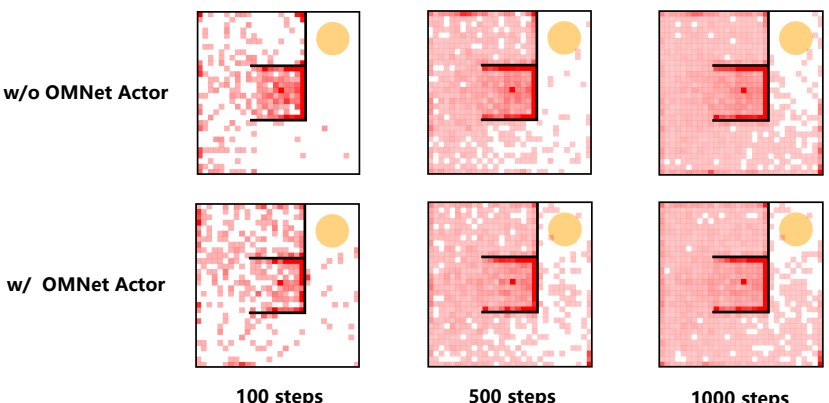

Figure 6: The visitation frequencies with and without using OMNet as the policy network.

In Figure 7, we visualize the trajectories rolled out by different subnetworks within OMNet as the policy network. All trajectories are obtained by rolling out in the evaluation environment using network parameters at selected time steps. We visualize the trajectories rolled out by all subnetworks after updating the policy network with different numbers of gradient steps. It can be seen that in the early stages of training (after 500 gradient updates), the trajectories generated by the five subnetworks exhibit high divergence. This suggests that we can obtain more diverse sample trajectories in the early stages of training by randomly selecting different subnetworks. Then, at step 4800, one of the subnetworks' evaluation trajectories successfully reaches the vicinity of the target position. Within just 200 gradient steps, the evaluation trajectories of all subnetworks have achieved success. This indicates that by maintaining multiple subnetworks to generate diverse trajectories when some subnetworks perform relatively better and obtain high-quality trajectory data earlier, other subnetworks can learn from these successful trajectories. In the subsequent training, the trajectories of subnetworks are optimized to reach the target point more quickly (6000 steps), and ultimately converge to nearly identical trajectories as training ends (30000 steps). In summary, these visual results illustrate how OMNet, acting as a policy network, samples more diverse trajectories to successfully enhance sampling efficiency.

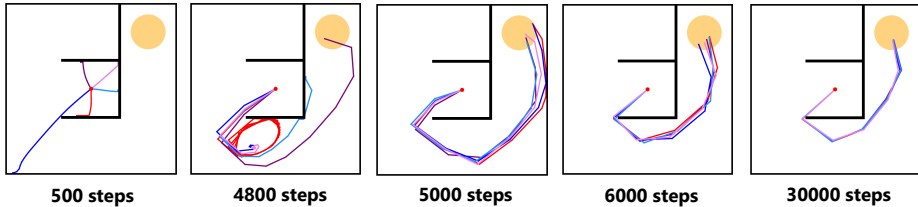

Figure 7: Trajectories sampled by different subnetworks within OMNet as the actor. We represent trajectories generated by different subnetworks using different colors.

## 4.4 OMNET ENABLES BETTER GENERALIZATION

The generalization performance of deep reinforcement learning algorithms is a critical consideration due to its implications for the practical applicability of these models in real-world scenarios (Rajeswaran et al., 2017; Packer et al., 2018; Kirk et al., 2023). In this section, we conduct experiments of OMNet in noisy environments. Note that this problem is directly relevant to real-world applications, such as the control of robots (Busoniu et al., 2006; Zhang et al., 2021; Höfer et al., 2021). These robots may need to learn from actual trajectories collected at different times and under varying conditions, each with its unique inherent biases.

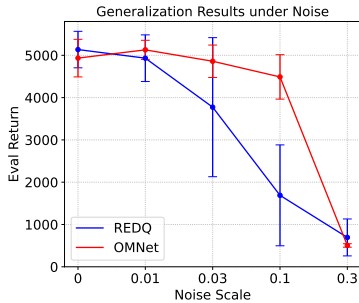

Figure 8: Generalization performance under different noise scale.

To compare whether OMNet can generalize better in a noisy environment compared to other algorithms, we utilize the Humanoid-v4 environment in MuJoCo and

introduce noise (as specified below) to the model's observations during both training and testing. For each episode, we sample an observation noise observation noise from a uniform distribution, i.e., $\sigma \sim \mathcal{U}[-\sigma_m, \sigma_m]$, where $\sigma_m$ represents the noise scale, determining the level of generalization difficulty. The noise remains constant throughout an episode. In Figure 8, we present curves illustrating the average return variation of REDQ and OMNet under varying levels of noise amplitude. It is evident that OMNet consistently maintains relatively high performance even when exposed to high levels of noise amplitude, demonstrating superior generalization capabilities compared to REDQ.

## 4.5 ABLATION STUDY

The implementation of OMNet is very straightforward and only requires two simple hyperparameters: the sparsity of subnetworks and the number of subnetworks. We conduct experiments with 10 different random seeds on all environments and present the results of the ablation study in Figure 9. From Figure 9(a), it can be observed that OMNet consistently achieves favorable performance within a within wide range of sparsity levels (Here the comparison is with the OMNet algorithm in Section 4.1, which uses a single value network). This demonstrates the robustness of OMNet with respect to the choice of sparsity values.

In Figure 9(b), we present the performance curves of OMNet under different numbers of subnetworks. OMNet exhibits strong stability across various subnetwork counts, even when using only 2 subnetworks, it achieves favorable performance. Furthermore, we observe that increasing the number of subnetworks (to 4 or 5) leads to improved algorithmic performance. Interestingly, we find that when we increase the number of subnetworks to 8, there is a slight drop in algorithmic scores. This suggests that the number of subnetworks may not be arbitrarily increased, and we delve further into this phenomenon in Section 4.6.

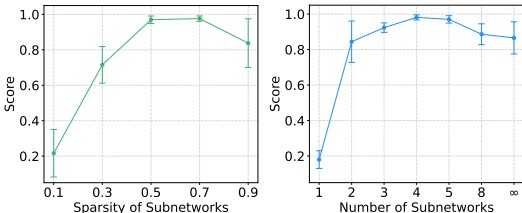

(a) Ablation of sparsity of subnetworks. (b) Ablation of number of subnetworks.

Figure 9: Results of the ablation experiments for OMNet. The scores are obtained by averaging the normalized evaluation returns across all environments (See Section 4.1).

## 4.6 "ONE IS INFINITY"?

From Figure 9(b), we observe that while increasing the number of subnetworks appropriately results in better performance than using only 2 subnetworks, this trend does not always hold. When we increased the number of subnetworks to 8, we noticed a decrease in performance compared to using 5 subnetworks. This sparked our curiosity about the case of even larger numbers of subnetworks.

Note that OMNet uses independently and identically distributed masks for each subnetwork, and we update network parameters or perform inference in a random manner. Thus, we can approximate the situation where the number of subnetworks tends to infinity by using randomly generated masks in each iteration or inference. This is similar to approaches such as Dropout (Srivastava et al., 2014) or DropConnect (Wan et al., 2013), generating a random mask dynamically during each forward process, as the former one using mask constrained on the neural level.

Through experiments, we found that using an infinite number of subnetworks, as shown in Figure 9(b), further reduced performance compared to using just 8 subnetworks, consistent with previous results. The performance curve reflects a trade-off: multiple subnetworks within one neural network can provide diverse outputs and greater effectiveness without extra parameters, but maintaining some distinction among subnetworks may be important. Using random, sparse masks dynamically may blur these distinctions. In summary, *one is more, but may not be infinity*. We believe this observation contributes to a better understanding of sparsity in neural networks and look forward to further research in this field.

## 5 CONCLUSION

In this paper, we investigate the possibility of discovering and leveraging multiple subnetworks within a single neural network. Our approach achieves more diversified outputs without increasing the parameter count and is empirically demonstrated to be effective and efficient in the field of deep reinforcement learning. We believe that the conclusions drawn in this paper contribute to a better understanding of the potential of neural networks and offer a perspective for more efficient training and inference.

REPRODUCIBILITY STATEMENT

In our paper, we provide a detailed description of the environments we tested and various baseline algorithms. We elaborate on the implementation of our algorithm in Section 3 and Appendix A, making it easily reproducible with simple modifications based on the baseline algorithms. Furthermore, Section 4.1 explicitly outlines the values of new hyperparameters used in our algorithm, while all other hyperparameters remain consistent with the default values used in the baseline algorithm. Our algorithm can be straightforwardly implemented and exhibits high reproducibility.

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

## A  TRAINING DETAILS OF SAC AND OMNET

In this section, we present the loss functions for SAC (Haarnoja et al., 2018) and OMNet, to illustrate how we conduct training based on the diversity of outputs from multiple subnetworks using a single neural network.

We first present the loss functions for SAC. SAC is an actor-critic RL algorithm, which requires training a policy network, i.e., $\pi_\phi(s)$, and two value networks which have the same architecture, i.e., $Q_{\theta^{(k)}}, k = 1, 2$. The loss functions used for SAC are:

$$
\begin{aligned}
\mathcal{L}_{\text{Critic}}(\theta^{(k)}) &= (Q_{\theta^{(k)}}(s_t, a_t) - (r + \gamma\hat{Q}(s_{t+1}, \pi_\phi(s_{t+1}))))^2, k = 1, 2, \\
\hat{Q}(s, a) &= \min\{Q_{\theta^{(1)}}(s, a), Q_{\theta^{(2)}}(s, a)\}, \\
\mathcal{L}_{\text{Actor}}(\phi) &= -\frac{1}{2}\sum_{k=1}^{2} Q_{\theta^{(k)}}(s_t, \pi_\phi(s_t, a_t)).
\end{aligned}
\tag{3}
$$

Note that SAC employs Clipped Double Q-Learning, as used in TD3 (Fujimoto et al., 2018). This approach has been proven effective in mitigating value overestimation issues in value-based learning.

With OMNet, we only need to maintain one policy network $\phi$ and one value network $\theta$. Denote $\theta_k$ as the parameter of the $k$-th subnetwork, the loss functions of critic is given by:

$$
\begin{aligned}
\mathcal{L}_{\text{Critic}}(\theta_i) &= (r + \gamma\hat{Q}(s_{t+1}, \pi_{\phi_{i''}}(s_{t+1})) - Q_{\theta_i}(s_t, a_t))^2, \quad i, i'' \sim \mathcal{U}[1, N], \\
\hat{Q}(s, a) &= \min\{Q_{\theta_{i'_1}}(s, a), Q_{\theta_{i'_2}}(s, a)\}, \quad i'_1, i'_2 \sim \mathcal{U}[1, N],
\end{aligned}
\tag{4}
$$

where $i$ is the randomly selected index of the subnetwork being updated in the critic, $i'_1$ and $i'_2$ are the randomly chosen subnetwork indices to calculate the Temporal-Difference (TD) target, and $i''$ is the subnetwork index used to determine the next-step action for the actor. In other words, we randomly select two from multiple subnetworks to calculate the TD target. This approach was inspired by Chen et al. (2021), but we efficiently perform this operation by utilizing multiple subnetworks within a single neural network, using much less parameters. The loss function of actor is given by

$$
\mathcal{L}_{\text{Actor}}(\phi_i) = -\frac{1}{N}\sum_{k=1}^{N} Q_{\theta_k}(s_t, \pi_{\phi_i}(s_t, a_t)), \quad i \sim \mathcal{U}[1, N],
\tag{5}
$$

where $i$ is the randomly selected index of the subnetwork being updated in the actor. We update the actor using the average value estimation of all subnetworks in the critic.

## B  DETAILS OF 2D MAZE ENVIRONMENT

### B.1  ENVIRONMENT SETTINGS

In this 1x1 maze shown in Figure 5(a), the agent always starts from the center of the maze, at the position $(1/2, 1/2)$. The agent's actions are two-dimensional vectors, and the range for each

dimension is $[-0.2, 0.2]$. The destination is located at the upper-right corner of the maze, at the position $(5/6, 5/6)$. When the agent reaches the vicinity of the destination (within a distance of less than 0.1) located in the top-right corner of the maze (the orange circle in Figure 5(a)), the game ends, and the agent receives a reward of $+100$. In all other cases, the agent does not receive any rewards. If the agent fails to reach the vicinity of the destination within 50 steps, the game is considered a failure. Three walls within the maze and walls surrounding the maze perimeter obstruct the agent from passing through.

### B.2 DETAILS REGARDING THE VISUALIZATION OF VISITATION FREQUENCY

We divide the entire map into a $30 \times 30$ grid and count the cumulative number of times the agent's position fell into each grid cell within a certain number of early exploration steps, converting it into a visit frequency. To ensure fair and reasonable comparisons, we conduct 10 repeated experiments and calculated the average results for comparison. Furthermore, since we aimed to compare the sampling policies generated by the policy network, we do not include the initial random exploration steps (1000 steps of warm-up) in the calculations. In Figure 6, we visualize the visit frequencies in different areas of the map, where areas marked in red indicate that the agent has reached that location, with darker colors representing higher visit frequencies.

## C DROQ AND SIN-DROQ WITH DIFFERENT DROPOUT RATES ON EACH ENVIRONMENT

Figure 10 and Figure 11 depict the training curves of DroQ and Sin-DroQ for different dropout rates in each environment. For DroQ, we adjusted the dropout rate in the range $\{0.001, 0.003, 0.01, 0.03, 0.1\}$. For Sin-DroQ, we adjusted the dropout rate in the range $\{0.001, 0.01, 0.1\}$. The selected optimal dropout rate for each environment can be found in Table 1. As observed, DroQ and Sin-DroQ exhibit sensitivity to the choice of dropout rate, with the optimal values varying significantly across different environments.

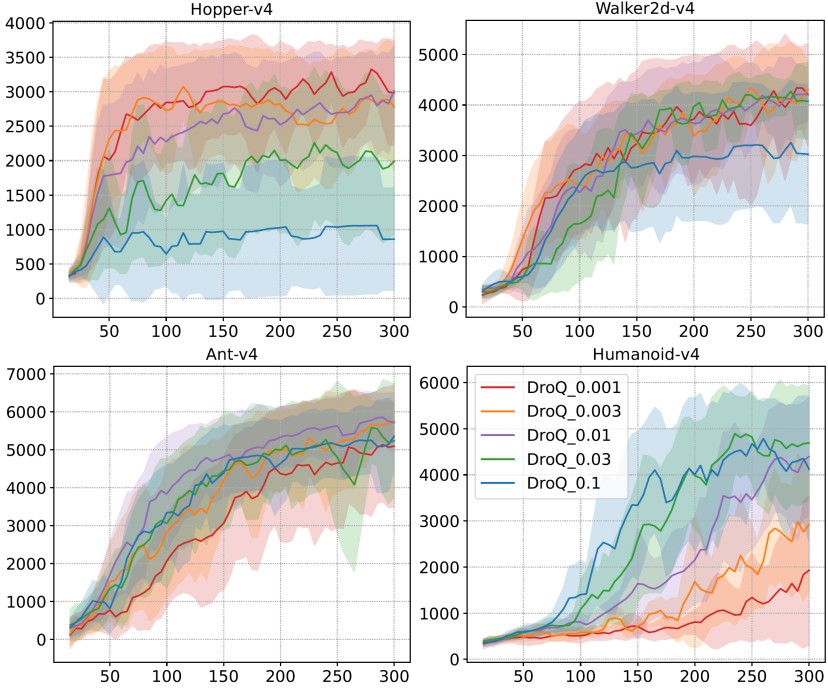

Figure 10: Training curves of DroQ with different dropout rates on different environments.

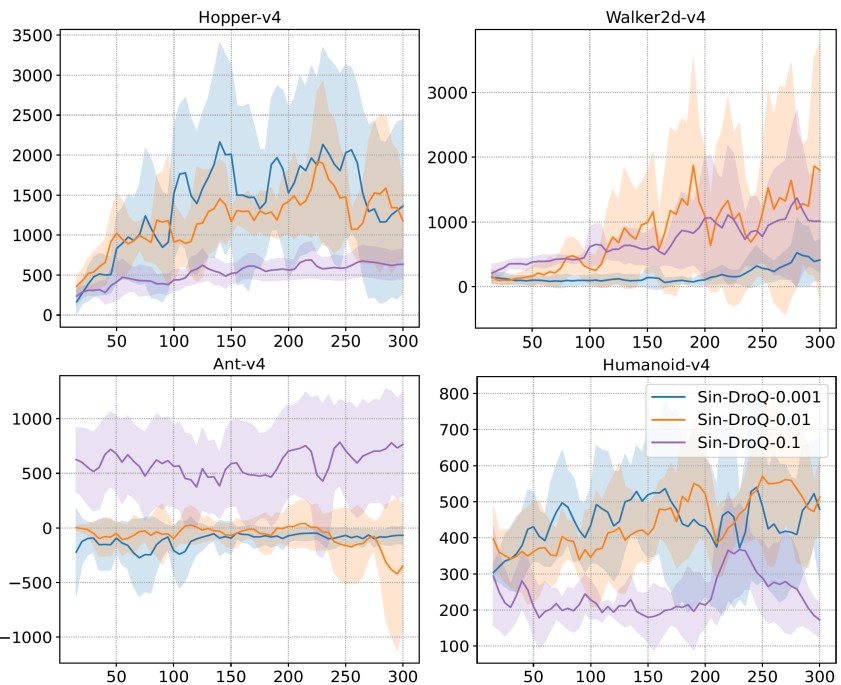

Figure 11: Training curves of Sin-DroQ with different dropout rates on different environments.

Table 1: Selected optimal dropout rate for DroQ and Sin-DroQ.

| Algorithm | Hopper-v4 | Walker2d-v4 | Ant-v4 | Humanoid-v4 |
|---|---|---|---|---|
| DroQ | 0.001 | 0.003 | 0.01 | 0.03 |
| Sin-DroQ | 0.001 | 0.01 | 0.1 | 0.01 |

