# OpenReview forum: "One is More: Diverse Perspectives within a Single Network for Efficient DRL"
_ICLR.cc/2024/Conference — ICLR 2024 Conference Withdrawn Submission_

### Official Review · Reviewer_9vY6 · 2023-10-31

**Soundness:** 3 good
**Presentation:** 3 good
**Contribution:** 2 fair
**Rating:** 6
**Confidence:** 3

**Summary:**

In this paper, the authors focus on the problem of sample efficiency in the context of Deep Reinforcement Learning. To this end, they propose One is More, a technique to leverage subnetworks within a single neural network to estimate value or policy. They evaluate their approach on continuous control environments, specifically on MuJoCo, demonstrating efficiency and computational advantages over baselines. Finally, they conduct an ablation analysis of their approach.

**Strengths:**

1. The approach is simple, intuitive, and makes conceptual sense. It is reasonable to assume that multiple subnetworks could potentially be leveraged within a single network.

2. While the choice of environments for evaluation is limited, the experimental analysis is thorough and convincing.

3. Quantitative results on the given environments is strong and shows promise.

**Weaknesses:**

1. The choice of environments for experimental evaluation is limited. Experiments are largely on a few tasks in MuJoCo and a maze. The paper's argument could be stronger if more complex environments were used (i.e. virtual robotics datasets such as Meta-World or Atari).

2. The implied analogy in Figure 1 is suspect. The connection between the OMNet subnetworks in an neural network trained for value/policy prediction and the structure of the human brain is weak.

**Questions:**

1. Do the experimental results on MuJoCo also generalize to other environments?

---

### Official Review · Reviewer_Y41k · 2023-11-01

**Soundness:** 2 fair
**Presentation:** 1 poor
**Contribution:** 2 fair
**Rating:** 3
**Confidence:** 5

**Summary:**

This paper presents OMNet, which utilizes multiple subnetworks within a single network, along with a set of binary masks for ensemble learning. The experimental results demonstrated improved performance on four tasks in MuJoCo.

**Strengths:**

1. The method is simple and can result in improved performance.

**Weaknesses:**

1. The presentation could benefit from eliminating redundant sentences, as they serve no essential purpose. I propose condensing the content preceding the experiments section into 2-3 pages.
2. The paper primarily focuses on a simple modification, and therefore, it is crucial to provide extensive experiments on how this straightforward technique can improve empirical performance. However, the paper's experimental scope is currently limited, encompassing only four tasks in MuJoCo and compares solely with SAC variants. To provide a more comprehensive assessment, the paper should extend its experiments to include a broader range of commonly used environments, including tasks in the Atari Learning Environment, DMControl Suite, DMLab, etc. Additionally, it should also compare the proposed method against a broader set of baseline algorithms, such as PPO, MPO, DDPG, TD3, and potentially DQN variants like Bootstrapped-DQN.
3. To demonstrate the effectiveness of this approach, the paper could also conduct experiments on offline reinforcement tasks, imitation learning tasks, and so on.

**Questions:**

1. Could the authors provide more experiments across different environments?

---

### Official Review · Reviewer_Vosg · 2023-11-07

**Soundness:** 2 fair
**Presentation:** 3 good
**Contribution:** 2 fair
**Rating:** 3
**Confidence:** 4

**Summary:**

This paper introduces a new method OMNet, which applies pre-computed binary masks on the parameters of the value and policy networks to create multiple sub-networks with high sample efficiency. OMNet is similar to dropout but its mask is not stochastic ones. The authors evaluate OMNet on four mujoco tasks Hopper-v4, Walker2d-v4, Ant-v4, and Humanoid-v4 to show the performance of OMNet is comparable to SOTA methods. It also shows that OMNet and REDQ do not suffer from value overestimation in Humanoid-v4. The suggested sparsity for the subnetworks is 0.5 and the number of subnetworks is 5.

**Strengths:**

- This paper introduces a simple yet effective algorithm that could improve the sample efficiency of conventional ensemble methods while achieving promising performance on continuous control tasks.

- The paper is well-written and easy to follow.

- The authors show interesting results not only justifying the stable performance of OMNet but also show the important property of avoiding value over-estimation, which would be important for replay-based approaches.

**Weaknesses:**

- As a technical paper for ICLR, I feel the method is a bit simple and lacks theoretical and technical novelty.

- The paper presents many interesting empirical results but I feel the evaluation domains are not very inclusive. It is insufficient to evaluate it only on four of the mujoco continuous control tasks, as similar works like Noisy Networks often use the Atari 2600 series for a thorough comparison. I feel the paper could be stronger if more continuous control tasks and discrete control tasks like Atari could be adopted. It is also unclear whether the method is capable of working on very complex (e.g., deep) value and policy networks, e.g., ResNet for IMPALA.

- Among the main learning curves presented in Figure 4(a), the performance of OMNet is on par with the SOTA methods, but not better.

- In many places of this paper, the authors criticize DropQ is sensitive to the dropout rate while claiming that OMNet's performance is STABLE across hyperparameter values, which I disagree. Figure 9(a) does show that OMNet's reward is fairly sensitive to the sparsity of subnetworks, which conflicts with the author's claim on OMNet's stability.

- For the 2D maze results shown in Figure 5, it is interesting to see that using OMNet in the policy network results in faster reward gaining. However, I feel there should be another baseline that removes OMNet from both the policy and value networks. I'm also curious would the methods eventually converge to the same standard?

- It is interesting to see that OMNet is better than REDQ in the noisy Humanoid-v4 environment with observation noise. I wonder if the finding is consistent in the other three tasks. DropQ should also be considered for generalization comparison.

**Questions:**

- What is the effect of using OMNet on purely policy-based methods without experience replay?  Will OMNet still preserve the property of more accurate value function estimation with diverse transitions?

- Is it possible to show the FPS (frames per second) for training and inference for the methods in Figure 3?

- Is the proposed method capable of working with large neural networks, e.g., resnets and LLAMA?

- Have you considered investigating the effect of using the mask on different types of layers, e.g., only on fc layers compared with on cnn or rnn layers?

- Do the points in Figure 9 correspond to the performance across all the four task domains? It would be good if the authors could include the sensitivity curve for each of the task in the appendix to see whether 5 subnets is an optimal choice for each of the games.

- Apart from Humanoid-v4, would SinSAC, SinDropQ, and SAC-20 also face over-estimation issues in the other three mujoco domains? It would be good if the value bias for the other three domains could be presented.

- The contribution of this paper would be significantly improved if the authors could combine OMNet with strong RL algorithms apart from SAC.

---

### Official Review · Reviewer_ZR2J · 2023-11-10

**Soundness:** 2 fair
**Presentation:** 2 fair
**Contribution:** 2 fair
**Rating:** 5
**Confidence:** 4

**Summary:**

The submission proposes to use a cost effective method for ensembling Q networks and policy networks in deep-RL.
To simulate multiple network predictions with a single network, binary parameter masks are randomly sampled and used to branch many different networks from a shared network.
Experimental results support that the proposed approach can achieve same effect of ensembling observed in the previous works.

**Strengths:**

The proposed method explores the use of a fixed set of parameter masks instead of dropout to simulate an ensemble of multiple Q networks , and as well as policy networks.
While previous work using dropout empirically requires at least two Q networks, the proposed approach can achieve same sample efficiency with only one network.
The idea is simple to implement but can roughly double the computational efficiency.

**Weaknesses:**

The comparison experiments with Sin-DroQ have some unclear aspects.
 - As far as I understood, DroQ baseline is M=2 case of DroQ and Sin-DroQ is M=1 case of DroQ (inferred from FLOPs in Figure 3). However, the details of this baseline is not explained in anywhere in the text, making it challenging for readers to follow. Considering the importance of this baseline, this lack of information is critical.
Additionally, the description of Sin-DroQ - "... the single-network version of DroQ, as originally proposed in..." - may be inaccurate as it seems DroQ uses M=2 in their experiments.
- The reason why Sin-DroQ fails is not discussed. Both DroQ and OMNet share a same idea of simulating multiple neural network with a single network, as dropout is well known to have a similar effect of training many different neural networks.
While the comparison results demonstrate DroQ empirically fails, and readers may be able to speculate about possible reasons why only OMNet is successful, the text does not explicitly discuss this core contribution of the idea.

Some of experimental results and their interpretation are also unclear:
- In Figure 4(b) and its interpretation in Section 4.2 - "... OMNet mainintaining a small value bias throughout the entire training process", it seems the low value bias is observed in the very beginning phase of the training and does not substantially lower than other baselines eventually. Also, this evaluation is done in only one environment, casting doubt on the claim of improved value bias.
- In Figure 5, the meaning plotted variances were not explained, and even the variances are very large and overlapped. Moreover, the qualitative visualization in Figure 6 does not show a significant difference. Therefore, the claim of diverse behavior of OMNet policy is not valid with the currently provided results.

The authors state their work is reproducible but did not include a detailed set of hyperparameters, such as learning rate or batch size.

**Questions:**

Despite having some disagreements with the claims in Section 4.2 and 4.3, the contribution of addressing the failure of Sin-DroQ could be substantial enough to share this work with the community.
Could the authors provide more intuitions for the failure of Sin-DroQ?